# Approximations to Diagnosis and Therapy of COVID-19 in Nervous Systems Using Extracellular Vesicles

**DOI:** 10.3390/pathogens11121501

**Published:** 2022-12-08

**Authors:** Karen Rojas, Maritza G. Verdugo-Molinares, Andrea G. Ochoa-Ruiz, Alejandro Canales, Edwin E. Reza-Zaldivar, Areli Limón-Rojas, Alba Adriana Vallejo-Cardona

**Affiliations:** 1Medical and Pharmaceutical Biotechnology Unit, Center for Research and Assistance in Technology and Design of the State of Jalisco, A.C. (CIATEJ), Av. Normalistas 800, Colinas de la Normal, Guadalajara C.P. 44270, Jalisco, Mexico; 2Tecnologico de Monterrey, Institute for Obesity Research, Av. General Ramón Corona 2514, Zapopan C.P. 45201, Jalisco, Mexico; 3Medical and Pharmaceutical Biotechnology Unit, CONACYT-Center for Research and Assistance in Technology and Design of the State of Jalisco, A.C., Av. Normalistas 800, Colinas de la Normal, Guadalajara C.P. 44270, Jalisco, Mexico

**Keywords:** extracellular vesicles (EVs), COVID-19, SARS-CoV-2, diagnostic, therapy

## Abstract

The SARS-CoV-2 virus was first identified at the end of December 2019, causing the disease known as COVID-19, which, due to the high degree of contagion, was declared a global pandemic as of 2020. The end of the isolation was in 2022, thanks to the global multidisciplinary work of the massive vaccination campaigns. Even with the current knowledge about this virus and the COVID-19 disease, there are many questions and challenges regarding diagnosis and therapy in the fight against this virus. One of the big problems is the so-called "long COVID", prolonged symptomatology characterized as a multiorgan disorder manifested as brain fog, fatigue, and shortness of breath, which persist chronically after the disease resolution. Therefore, this review proposes using extracellular vesicles (EVs) as a therapeutic or diagnostic option to confront the sequelae of the disease at the central nervous system level. Development: the review of updated knowledge about SARS-CoV-2 and COVID-19 is generally addressed as well as the current classification of extracellular vesicles and their proposed use in therapy and diagnosis. Through an analysis of examples, extracellular vesicles are highlighted to learn what happens in the central nervous system during and after COVID-19 and as a therapeutic option. Conclusions: even though there are limitations in the knowledge of the neurological manifestations of COVID-19, it is possible to observe the potential use of extracellular vesicles in therapy or as a diagnostic method and even the importance of their study for the knowledge of the pathophysiology of the disease

## 1. Introduction

The central nervous system (CNS) can be affected after a viral infection, producing mild neurological manifestations such as headaches and even severe conditions such as encephalitis. It has been proved that some viruses, such as influenza virus, respiratory syncytial virus, enterovirus D68, HIV, and the current pandemic virus SARS-CoV-2, are neurotropic [1].

In the study of the pathogenesis of viral diseases, scientists have been a significant interest in the study of extracellular vesicles whose participation in the spread of the disease has been confirmed. Viruses are significantly similar to EVs in their structure, size, generation, and absorption. Cells infected by viruses release a content of EVs that arouse the host's immune response, but not always positively. In some cases, the EVs can transfer the same components of the virus and thus facilitate its spread or allow it to spread without being detected by the immune system. HIV, hepatitis C, herpesviruses, cytomegalovirus, coronavirus, and SARS-CoV-2 have been related to EVs, modulating their spread [2]. 

The main access route of SARS-CoV-2 to infect cells is through the angiotensin-converting enzyme 2 receptors (ACE2). Therefore, this receptor has been widely studied since the first appearance of SARS-CoV. ACE2 is present in the lungs and other organs such as the stomach, small intestine, colon, skin, spinal cord, spleen, liver, kidneys, and brain [2,3]. This multiorgan ACE2 expression could explain the variety of symptoms during the COVID-19 progression and after months of the disease resolution. Besides typical COVID-19 symptoms, there are specific neurological manifestations such as poor concentration, headaches, depression, or neuropsychiatric syndromes, giving origin to the term "long COVID" [4]. All these neurological sequelae are probably due to the SARS-CoV-2 can infect glial cells and neurons in the brainstem, the paraventricular nucleus, nucleus tractus solitarius, and the rostral ventrolateral medulla, and hippocampus, promoting the inflammatory markers IL-6, IL-1, CKAP4, and GAL-9 overproduction, glial and microglia activation [5,6] as well as the neuronal degeneration in the hippocampus [7]. The most critical requirements to face this pandemic are the diagnosis, treatment, prevention of the infection and its acute symptoms, and the development of treatments aimed at the central and peripheral nervous system to mitigate the long COVID manifestations. 

In this review, we explored the relationship of SARS-CoV-2 with the CNS. We proposed extracellular vesicles (EVs) as an option for therapy or detecting long-COVID-19 neurologic sequelae. In the detection context, EVs contain numerous key molecules for cell communication by secreting proteins, RNA, carbohydrates, and lipids, among others, allowing the transfer of information for the disease's progression. EVs influence target cells through receptor-ligand interaction, direct fusion, and internalization [2]. Moreover, they could be considered suitable biomarkers or biosensors due to the EV transport in different fluids [8].

Proposing EVs as therapy is based on reports of liposomes as an alternative for drug delivery due to their flexibility, stability, and the possibility of programming drug dosage according to the bilayer composition [9]. On the other hand, it has been reported that EVs play an essential role in the communication mechanism. Due to their composition and availability in different biological fluids, they could be considered suitable biomarkers or biosensors. EVs are also analyzed as an alternative to transporting therapeutic molecules. It should be noted that EVs tend to undergo changes in both their composition and concentration in circulation when specific pathologies occur [8]. Also, EVs have a great potential to be used on CNS because they reflect the status of the disease and be able to pass the blood-brain barrier to the rest of the body and vice versa [10]. 

## 2. Central Nervous System and COVID-19

SARS-CoV-2 is considered a respiratory virus whose main route of infection is through the inhalation of aerosols or contaminated saliva droplets dispersed in the air [11,12]. COVID-19 is the pathology caused by the SARS-CoV-2 virus. Worldwide, this disease has registered around 600 million infections and more than 6.5 million deaths until January 2022 [13].

Like other coronaviruses, it binds to the ACE2 receptor to infect cells. After attaching to ACE2, the spike protein is cleaved by transmembrane serine protease 2 (TMPRSS2) to allow subsequent virus entry into the cell [14]. Other host proteins may allow the virus internalization, including neuropilin-1 (NRP-1), cathepsin, and basigin. Consequently, the cell type with the highest frequency of infection present includes the pneumocytes of the lower respiratory tract, being the primary target, the vascular endothelium cells, kidney, muscle, and brain [11,12,15]. The most severe effects of COVID-19 include damage to blood vessels, the brain, gastrointestinal tract, kidneys, heart, and liver [11].

Different clinical and experimental studies have provided ample evidence that SARS-CoV-2 has a tropism for the central nervous system (CNS). This type of virus remains in the respiratory tract. It has also been shown to invade the CNS, affecting neurons and glial cells, producing severe neurological alterations and inducing neurological diseases [1,6,7]. Therefore, a neurological symptom could be the first manifestation of COVID-19. However, further complications associated with COVID-19 are still unknown. More than 50% of hospitalized patients have presented neurological symptoms compared with non-hospitalized patients or milder manifestations of the disease [11,12]. Research conducted in Wuhan, China, found that 36% of COVID-19 patients suffered from neurological symptoms, of which 25% may be due to direct invasion of the brain and spinal cord [16]. Patients who presented a severe COVID-19 infection have developed more neurological manifestations [17,18]. 

The most common neurological symptoms reported by COVID-19 include headache, dizziness, malaise, anosmia and hypogeusia, seizures, encephalopathy, acute cerebrovascular disease, epilepsy, psychiatric disorders, encephalitis, neuropathies, stroke, and other manifestations associated with the peripheral nervous system [11]. Headache has been the most prevalent symptom in affected individuals. For the reason that the olfactory epithelium is the main route of entry of the virus into the body, anosmia followed by ageusia are the most frequent manifestations of SARS-CoV-2 infection. The loss of smell is due to many ACE2 receptors present in the ciliated and goblet cells of the nasal epithelium, causing it to be a reservoir of the virus, associated with a greater susceptibility to infection [11,12,15,19]. Similarly, according to a study that included 3148 patients, the most frequently developed symptoms were headaches, olfactory dysfunction, and taste disorders. Olfactory dysfunction was inversely related to patient age. In older patients, the prevalence of this symptom decreased [1].

Headache has been the most prevalent symptom in affected individuals. It is considered one of the long-term conditions of COVID-19 (“long COVID”), presenting this symptom in 44% of 80% of the population analyzed, followed by fatigue with 58% and disorder care in 27%, which indicates that the post-infection effects that prevail are those associated with the nervous system, according to a meta-analysis study [20]. This may be because a large proportion of ACE2 receptors have been found on vascular smooth muscle cells in the brain, liver, spleen, kidney, oral mucosa, small intestine (ileum), and colon. Where specifically in the brain, the receptors are found in glial cells, neurons, and cerebrospinal fluid [4,7,9,12,16,21]

Although the exact route by which SARS-CoV-2 invades the CNS is still unknown, different mechanisms have been proposed to understand these neurological conditions, as shown in the schematic of Figure 1.

Some proposed theories are the following: (1)Direct virus invasion, the virus directly invades the nervous tissue because it contains a considerable number of ACE2 receptors and can cross the various barriers of the nervous system.(2)Inflammation by cytokine storm, the neurological damage is the product of the inflammation of the cytosine storm caused by the systemic infection of SARS-CoV-2. The cytokines can reach the hypothalamus and consequently increase the patient's neuroinflammation and corporal temperature, promoting fever [11,19]. Alternatively, the cytokine storm can cause the breakdown of the BBB and allow monocytes to traverse the BBB physically.(3)Taste cells pathway, the virus or an inflammatory process can directly affect the taste cells by its high content of ACE2 receptors in the basal layer of the olfactory bulb epithelium, the vascular endothelium, and in vascular smooth muscle cells [4,7].(4)Nasal mucosa pathway, recent studies have detected SARS-CoV-2 RNA particles in the olfactory mucosa and in areas of the brain that receive projections from the olfactory tract. Therefore, it has been considered that there may be a virus infection by axonal transport [4,11].

The CNS vulnerability to SARS-CoV-2 infection probably includes different receptors such as basigin (BSG, CD147), neuropilin 1 (NRP-1), and the action of proteases such as cathepsin B and L and furin. Norepinephrine, glucocorticoids, and calprotectin are released during lung damage and stimulate a process in the bone marrow (emergency myelopoiesis), which favors the formation of granulocytes and reduces the formation of lymphocytes, causing lymphopenia and neutrophilia that are prevalent features of COVID-19. It has been shown that the SARS-CoV-2 infection of the CNS can lead to encephalopathy. The immune response can trigger an autoimmune reaction to the myelin sheath, causing demyelination of some areas of the nervous system. At the CNS level, this causes acute disseminated encephalomyelitis. Viral respiratory tract infections have been associated with neurological and psychiatric sequelae, including dementia, depression, post-traumatic stress disorder, and anxiety [11,12,15].

Most manifestations occur early in the disease [22]. A recent study showed that at 30 days of follow-up, the most frequent symptom was headache; at 90 days of follow-up, the most frequent symptom was insomnia [17]. However, some symptoms persisted for 3 to 9 months after diagnosis [18]. Likewise, another study using a US data network revealed that 33.62% showed neurological symptoms six months after the diagnosis of COVID-19 [18]. Manifestations such as anxiety, depression, fatigue, or muscle weakness are also reported after six months of infection [11].

For this reason, the term "long COVID" has been associated with patients presenting severe symptoms of COVID-19, including long-term neurological and psychiatric sequelae. Biomarkers associated with astrocyte and neuron damage, such as NfL and GFAP, increased depending on the severity of COVID-19. Even though these biomarkers normalized six months after infection, the patients manifested fatigue and decreased cognitive activities and attention [23]. The physiopathology of post-COVID-19 effects is still unknown. A study at the University de Sant Joan de Reus tried to recollect information about the diminution of headaches in patients after vaccination against COVID-19; even with the limitations, the hypothesis of the diminution of neurological cannot be discarded [24]. Sporadic cases of neurological effects after vaccination for Anti-LGI1 Encephalitis [19], Guillain-Barre syndrome (GBS) [25], and Neuralgic amyotrophy [26] have been reported, highlighting that the vaccine may influence CNS. Frontera et al. made a recompilation of data to analyze rates of neurological events following vaccines. This analysis is interesting because the vaccine that uses liposomes has a low rate of neurological effects [27], which could be related to using liposomes, ARNm, or both.

Although common neurologic symptoms include cognitive or memory disorders, headache, loss of smell or taste, and myalgia, more severe illnesses such as encephalopathy, delirium, cerebrovascular disease, seizures, neuropathy, and myopathy have occurred, and some less frequently reported include abnormal movements, psychomotor agitation, syncope, and autonomic dysfunction [22]. Neurological symptoms may be associated with fatal diseases. Neuroimaging analyses revealed that ischemic cerebrovascular accidents, leptomeningeal enhancement, and encephalitis were the most frequent. Confusion was the most frequent neurological manifestation, followed by altered consciousness and clinical signs of corticospinal tract involvement, agitation, and headache [28].

## 3. Extracellular Vesicles in the Central Nervous System

EVs are membrane particles involved in regulating physiological and pathological processes. The ability to mediate cellular communication makes them critical players in cellular processes [29].

Virtually all cells release EVs under normal conditions, but their release and composition can be altered in pathological conditions, making them a potential biomarker of various diseases. The concentration of EVs can be measured in biological fluids such as blood, saliva, urine, cerebrospinal fluid (CSF), semen, and bile [30]. Their composition includes proteins, peptides, micro-RNA, and nucleic acids, which mediate cellular signaling cascades of damage or protective mechanisms and help to dispose of toxic cellular products [31]. Different EV types have specific characteristics; in Table 1, there is summarized information reviewed.

Exosomes, the smallest subclass of EVs, contain molecules provided by the Golgi apparatus, endocytosis, and the cytoplasm. On the surface of exosomes, there is a receptor-ligand interaction [33]. Microvesicles sprouting directly from the plasma membrane have been characterized as having important biomarkers since these microvesicles can be released in response to a specific stimulus or condition [2].

EVs play crucial roles in various pathologies. They can act as disease progression agents, from cancer to neurodegenerative diseases. Multiple studies proposed using EVs to diagnose or treat diseases with promising results [34,35]. In a cerebral pathology, the change in the lipid profile of the EVs was reported. The pathology studied was Alzheimer’s. The alteration of phosphatidylcholine (PC) and ethanolamine (PE) was reported, as well as the function of phosphatidyl serine (PS) and alkyl and alkylene esters (plasmalogen) that are associated with the uptake of EVs into cells [36].

In a study carried out by Vacchi et al., a diagnostic model based on extracellular vesicles was developed for the potential differentiation between Parkinson's disease and atypical parkinsonism by identifying differentially expressed antigens related to the severity of the disease [37].

In anticancer research, EVs mediate cell communication within a tumor environment and induce phenotypic modification. The effects of the EVs content and their roles in tumorigenesis and metastasis have been investigated. For example, microRNAs from EVs secreted by colorectal cancer cell lines were tested to predict colorectal cancer. A correlation analysis was carried out between the microRNA profiles of EVs derived from cell lines and tumor tissues, revealing a high sensitivity (94.9%) and specificity (100%) to predict tumors and identify miR-7641 as a potential biomarker for diagnosis [38]. Likewise, it has been observed that EVs from osteosarcoma can regulate the hypomethylated DNA of LINE1, causing epigenetic changes [39].

Similarly, Lo et al. reported that blood samples from patients diagnosed with amyotrophic lateral sclerosis have more prominent vesicles than control patients. Moreover, they identified microRNA signatures that correlated with the pathology, reflecting the poor health status of the patient [40].

EVs have also been involved in diseases caused by pathogens. It has been shown how viruses and bacteria can alter EVs synthesis to improve its propagation. The EVs released from infected cells reveal mechanisms of interactions between the pathogen and the host cell, such as replication and regulation of the response of the host immune system. To an effective infection, viruses seek an optimization of viral propagation, which EVs can mediate. The role of EVs in viral replication includes the mechanism by which viruses enter the cell and the ability to evade the host's immune response. Non-enveloped viruses use EVs to get out of cells, avoiding the immune system response. Conversely, enveloped viruses that use EVs as a vehicle to enhance their viral propagation.

In conclusion, viruses take advantage of EVs. Both enveloped and non-enveloped viruses use EVs to optimize their viral replication. They could use them to evade the immune system's response and thus avoid being eliminated and carry out viral transfer to healthy cells [41,42].

EVs also play a fundamental role in the CNS maintaining cellular homeostasis by regulating the cleaning of protein aggregates, pathogenic agents, or any unwanted biomolecule. However, they also can transfer harmful substances to healthy nerve cells in pathological conditions. EVs composition provides critical information about neurological diseases, so their role as biomarkers could be promising. Similarly, their role as therapeutic agents is interesting because EVs can pass the blood-brain barrier (BBB) and be exciting vectors for drug delivery systems for brain diseases [28,43,44].

Neuronal-enriched EVs (nEVs) have been characterized to understand the mechanisms of inflammation and neurological dysfunction in post-COVID-19 recovery. In a study with 24 patients, even though only 8 showed neurological symptoms, all had the presence of nEVs in plasma with an average size of 75–125 nm regardless of the time post-infection. Within the nEVs, neurodegenerative proteins such as HMGB1 and inflammatory cytokines were found; the most remarkable is IL-4, which is related to neurological functions such as memory [31].

## 4. Liposomes and Vesicles for Treatment of COVID-19 in the Central Nervous System

The first vaccine for the prevention of COVID-19 to receive authorization from the health authorities was BNT162b2, developed by the Pfizer-BioNTech laboratories. This vaccine has exciting features, as it is the first approved vaccine that uses mRNA as part of the vaccine mechanism. This mRNA codes for the spike protein of the virus and is protected in a liposomal structure. When the liposome encounters the cells, it delivers this genetic material, which is deposited in the cytoplasm, where the ribosomes are responsible for transcribing, causing the cell to express the spike protein of the virus. Once the protein is expressed, antibodies have generated that alert the immune system against SARS-CoV-2. It is still unknown how long the immunity induced is, but it is estimated that it could be between 6 to 9 months [45].

Bansal and collaborators (2021) carried out an experiment in which they showed that the vaccine induces the production of exosomes that express the spike protein of the virus, representing a novel activation of the immunity induced by mRNA vaccines. As part of the experiment, mice were immunized with isolated exosomes. The results showed that exosomes with the spike protein of the virus are immunogenic. Therefore, exosomes could be proposed as part of the mechanism of action of mRNA vaccines [46]. Exosomes pass through BBB and establish communication between the CNS and the rest of the body [47]. This capacity leaves gaps regarding the implications of the immunogenic exosomes in vaccination.

Since ACE2 expression is directly associated with SARS-CoV-2 infection, a study compared the ACE2 expression in plasma extracellular vesicles of COVID-19 patients. In the acute phase of the disease, a more significant amount of ACE2 is expressed in extracellular vesicles. In addition to this observation, it was found that these vesicles can neutralize the virus [48].

Another interesting point of view is the one raised in the review by Lam, Huang & Shui (2022), which states that lipid metabolism is one of the pathways altered during SARS-CoV-2 infection. Positive RNA viruses such as this tend to control the lipid membrane of the host cell as part of their viral replication. Inhibition of these pathways also decreases virus replication. Coronaviruses are also involved in the maturation of multivesicular bodies, as they can introduce viral content into EVs [49].

Liposomes and EVs' ability to contain mRNA and transmembrane proteins have led to various investigations. In a pilot test, EVs expressing ACE2 alone or in combination with TMPRSS2 prevent lentivirus infection expressing the SARS-CoV-2 spike protein in human cell lines lung epithelium (Calu-3), epithelial cells of colorectal cancer (Caco-2) and genetically modified 293FT. Infection decrease was correlated with EVs concentration. EVs expressing only ACE2 or in combination with TMPRSS2 reduce the entry of the pseudovirus into the cell lines. Therefore, engineered EVs could be a therapeutic option for preventing COVID-19 [14].

Exosomes allow the restoration of neurological abnormalities by suppressing neuronal apoptosis and promoting functional recovery. Being the smallest subclass of EVs, they can enter the blood tissue barrier and repair injured neurons during viral infection. Likewise, exosomes promote endogenous repair and reduce the cytokine storm [50]. Using exosomes derived from mesenchymal stem cells showed effectiveness in a clinical study. An intravenous dose of exosomes in COVID-19 critical patients registered a decrease in hypoxia, improved immune response, and decreased cytokine storm [51].

Even if exosomes could be used as important biomarkers for disease diagnosis, no reports of COVID-19 diagnosis have been found. Diagnostic approaches are still based on antibody-antigen recognition of SARS-CoV-2. Abubakar and his team developed an electrochemical immunosensor for detecting the virus through the electrodeposition of graphene and gold in glassy carbon electrodes. When tested on patients diagnosed with COVID-19, effective screening was demonstrated with a sample for low detection, so its compact, portable, and effective use will allow rapid diagnosis [52]. Figure 2 schematizes the change in composition and structure of the exosomes that they undergo post-infection and their possible use in therapy, diagnosis, or detection of infection in so-called long covid.

## 5. Conclusions

The COVID-19 pandemic has brought countless complications, not only at the respiratory level but also at the CNS level. Thus, the development of effective treatments for long COVID-19 neurological sequelae is required. It is well known that more options of minimally invasive molecules are needed for patients to treat the disease's adverse effects.

EVs present receptors that facilitate the infection and spread of the COVID-19 virus, so using them as a tool for diagnosis or treatment has caught the attention of researchers. Their biocompatibility and nanometric size are some of the advantages that the EVs make them very suitable candidates to be used as a tool for diagnosis. Likewise, it has been demonstrated that they have performed satisfactorily for the delivery of drugs and that due to their size and being able to enter the cerebrospinal fluid barrier is possible to attack the problem at the neuronal level. However, it is necessary to expand the number of investigations to vanish the obstacles they present and clarify how the virus affects the brain.

One limitation to developing a therapy for neurological manifestations of COVID-19 is that the mechanism of the virus in the brain is not yet known with certainty. However, developing vaccines and effective treatments for the disease could be the beginning of reducing the consequences observed in the Central Nervous System. The use of liposomes and EVs is promising for these purposes, and given their characteristics, they could be helpful for both requirements: obtaining biomarkers and developing treatments. The role of EVs in the immunology of the disease, and probably in its pathogenesis, could also give them an essential role as a therapeutic target.

Likewise, to enhance knowledge of EVs for their application in the treatment or diagnosis of long covid, both in silico and experimental strategies can be designed to elucidate the possible compositional and structural changes that EVs could present post-infection, considering the symptoms that affect the nervous system in the short and long term.

In silico tools and omics can be helpful to find those changes at the RNA, DNA, protein, or lipid level that converge to determine the periodicity of the post-infection condition.

## Figures and Tables

**Figure 1 pathogens-11-01501-f001:**
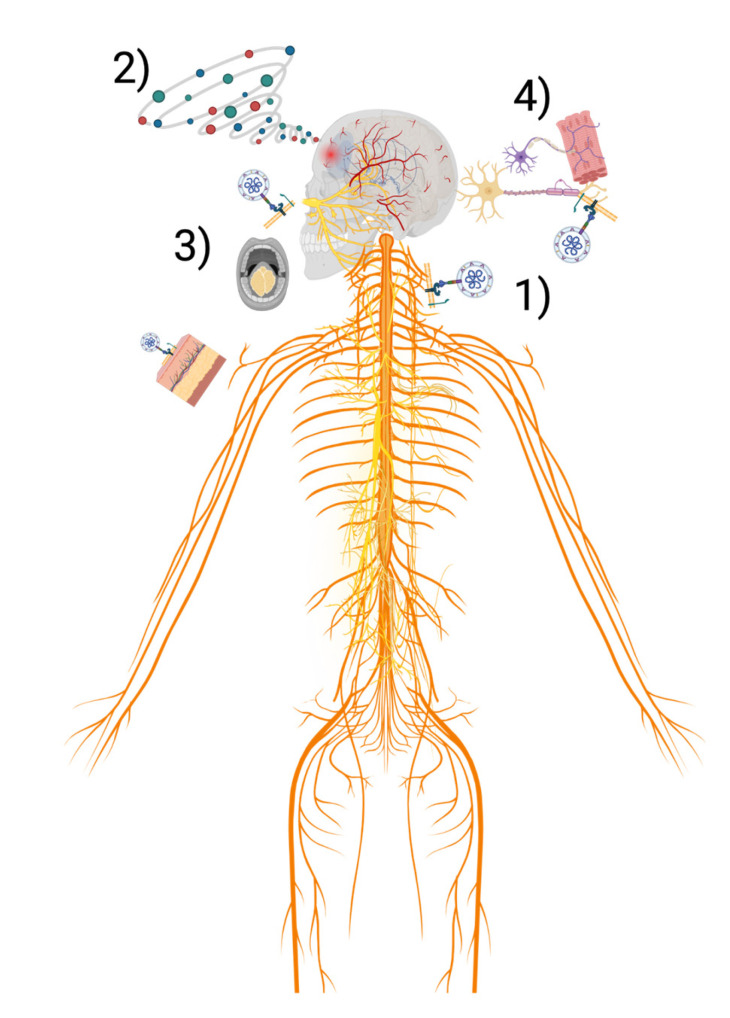
Proposed mechanisms by which SARS-CoV-2 invades CNS (1) Direct virus invasion of nervous tissue (2) Inflammation by cytokine storm (3) Taste cells pathway (4) Nasal mucosa pathway. Created with BioRender.com.

**Figure 2 pathogens-11-01501-f002:**
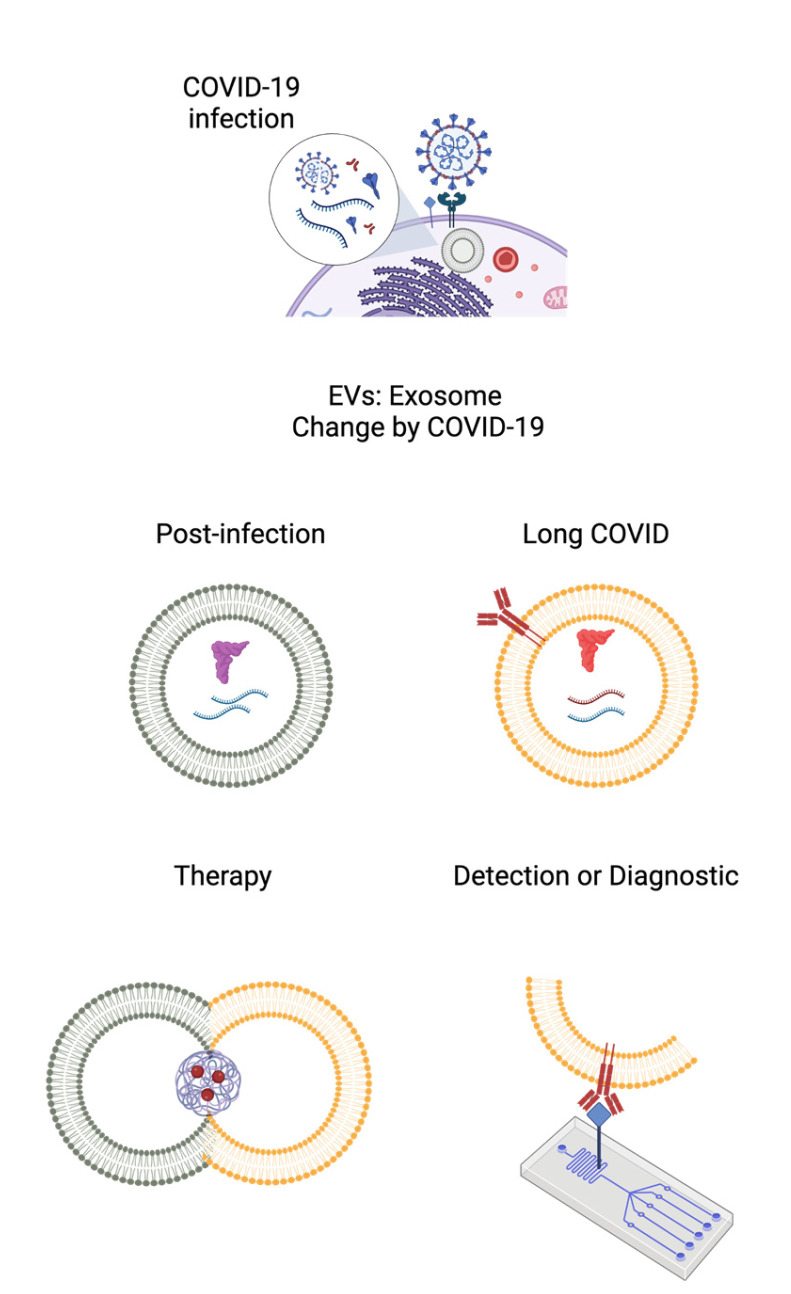
Scheme of the use of exosomes for the treatment and diagnosis or detection of "Covid Long". Created with BioRender.com.

**Table 1 pathogens-11-01501-t001:** Subtypes of EVs [32].

Subtype of EVs	Diameter (nm)	Releasing Pathway	Biogenesis
Exosomes	30–150	Exocytosis results from multivesicular bodies (MVBs) and plasma membrane (PM) fusion.	Via a ceramide or endosomal sorting complex required for transport (ESCRT)-dependent pathway.
Microvesicles	100–1000	Outward protrusion or budding of PM.	Includes calcium-dependent signaling, cytoskeleton remodeling, and externalization of phosphatidylserine (PS) with phosphatidylethanolamine (PE). The complete mechanism of membrane changes and signaling is still under investigation.
Apoptotic bodies	50–5000	Apoptotic cells.	During apoptosis.

## Data Availability

Not applicable.

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
