# Peer review of "Approximations to Diagnosis and Therapy of COVID-19 in Nervous Systems Using Extracellular Vesicles"

_pathogens, 2022, doi:10.3390/pathogens11121501_

Round 1
Reviewer 1 Report
Title: Approximations to diagnosis and Therapy of Covid-19 in 2 Nervous Systems Using Extracellular Vesicles
Manuscript Number: pathogens-2011279
In this manuscript authors have claimed about SARS-CoV-2 and COVID-19 is generally 26 addressed, as well as the current classification of extracellular vesicles and their proposed use in 27 therapy and diagnosis. This is interesting work. However, there are some major revisions for better understanding and improvement of manuscript as below:
1. Several latest literatures are available on same studies authors have cited very few references in introduction part. So, introduction part should need to be extend with latest references.
2. Figure 1 is very basic, authors what they want give the information to audience. Should be improve scientifically.
3. Table 1, reference citation is required.
4. Already several articles are reported on the same studies but authors have been used only one figure in current manuscript. Author should take the permission from other reported work for more figure.
5. Authors should be explaining the novelty of work because several articles reported as:
doi: 10.1016/j.meegid.2020.104422
doi: 10.3390/biomedicines9101373
https://doi.org/10.3389/fmolb.2021.699929
6. Conclusion part is very poor no futuristic explanation, need to revised.
7. Authors should check typographical and grammatical mistakes in chapter.
8. Need to be cite some relevant references for better presentation of manuscript.
https://doi.org/10.1021/acsabm.2c00301
DOI https://doi.org/10.1039/D2TB01409B
DOI: 10.1016/j.matlet.2021.130898
Advanced Biosensors for Virus Detection: Smart Diagnostics to Combat against SARS-CoV2 Pandemic, 2022. Elsevier, Paperback ISBN: 9780128244944
Computational approaches for novel therapeutic and diagnostic designing to mitigate SARS-CoV2 infection”. 2022. Elsevier, Paperback ISBN: 9780323911726
Author Response
Dear Revisor,
First, we appreciate the time you spent reviewing the writing, as well as your comments, which we respond to. We made the changes, and we highlighted them in the text in yellow.
- Several latest literatures are available on same studies authors have cited very few references in introduction part. So, introduction part should need to be extend with latest references.
We listened to the suggestion and added the information to the text, highlighting the changes in yellow.
- Figure 1 is very basic, authors what they want give the information to audience. Should be improve scientifically.
We changed Figure 1 and expanded the explanation on the text.
- Table 1, reference citation is required.
Add the reference to the table.
- Already several articles are reported on the same studies but authors have been used only one figure in current manuscript. Author should take the permission from other reported work for more figure.
We attach a second image to exemplify the use of EVs as treatment and diagnosis, based on the studies available for the treatment with other conditions.
- Authors should be explaining the novelty of work because several articles reported as:
In the cited reports, interesting data can be obtained regarding the participation of the EVs during the infection, and their participation in transmitting the infection due to a change in their structure during the infection. They also condense information for the use of EVs as therapeutic agents through drug delivery, as vehicles for vaccines and the promotion of cell recovery and proliferation using stem cells and EVs. They also report the pros and cons of the use of EVs and the need to continue with the study on their extraction and study of function and composition.
Therefore, the articles were cited in different sections of the manuscript, and are used as a basis for proposing design strategies for therapy and diagnosis in the sequelae of long covid present in the nervous system. (doi: 10.1016/j.meegid.2020.104422, doi: 10.3390/biomedicines9101373, doi: 10.3389/fmolb.2021.699929)
- Conclusion part is very poor no futuristic explanation, need to revised.
We listened to the suggestions for the conclusions and added and some opportunity strategies for the use of EVs, however there is still a long way to go to be able for identify the changes in EVs during the long covid, which opens a window of opportunity for various investigations and developments.
- Authors should check typographical and grammatical mistakes in chapter.
Corrected typographical and grammatical errors.
- Need to be cite some relevant references for better presentation of manuscript.
In the conclusions we consider the literature proposed for POC. And we add other articles to complement the review work, maintaining the proposal of the importance of EVs to be considered an alternative therapy and diagnosis in the symptoms that occur in long Covid.
We again appreciate your suggestions, and we hope that with the changes, the review will be of interest, to promote the study of EVs in the central nervous system in chronic pathologies such as long covid.
Sincerely
Alba Adriana Vallejo Cardona
Reviewer 2 Report
The topic of the work is very current and useful regarding the display of the latest diagnostic and therapeutic possibilities in the neurological manifestations that can occur during COVID-19 infection or as part of "long COVID" syndrome.
The authors have gathered the latest knowledge from relevant literary sources.
The conclusion clearly highlights the place and role of extracellular vesicles in the neurological manifestations of COVID-19 infection and opens topics for further research, especially regarding vaccines.
Author Response
We thank the reviewer for his comments and mention that some changes were made to the text in response to the suggestions of reviewer 1
Best regards.
Alba Adriana Vallejo Cardona
Round 2
Reviewer 1 Report
The authors have addressed the all comments very nicely. The current form may be acceptable for publication.